# *Toxoplasma gondii* infection and insomnia: A case control seroprevalence study

**Cosme Alvarado-Esquivel** [iD] [1]*, **Sergio Estrada-Martínez**[2], **Alma Rosa Pérez-Álamos**[2],
**Agar Ramos-Nevárez**[3], **Karina Botello-Calderón**[1], **Ángel Osvaldo Alvarado-Félix**[1],
**Raquel Vaquera-Enríquez**[4], **Gustavo Alexis Alvarado-Félix**[1], **Antonio Sifuentes-Álvarez**[1],
**Carlos Alberto Guido-Arreola**[3], **Elizabeth Rábago-Sánchez**[1], **Leandro Saenz-Soto**[3]

**1** Biomedical Research Laboratory, Faculty of Medicine and Nutrition, Juárez University of Durango State,
Durango, Mexico, **2** Institute for Scientific Research "Dr. Roberto Rivera-Damm", Juárez University of
Durango State, Durango, Mexico, **3** Clínica de Medicina Familiar, Instituto de Seguridad y Servicios Sociales
de los Trabajadores del Estado, Durango, Mexico, **4** Health Center No. 2 "Dr. Carlos Santamaría", Servicios
de Salud de Durango, Durango, Durango, Mexico

* alvaradocosme@yahoo.com

pone.0266214

Medical Sciences, ISLAMIC REPUBLIC OF IRAN

**Data Availability Statement:** All relevant data are
within the paper and its Supporting Information
file.

## Abstract

We determined the association between *Toxoplasma gondii* (*T. gondii*) infection and insom-
nia. Through an age-and gender-matched case-control study, 577 people with insomnia
(cases) and 577 people without insomnia (controls) were tested for anti-*T. gondii* IgG and
IgM antibodies using commercially available enzyme-immunoassays. Anti-*T. gondii* IgG
antibodies were found in 71 (12.3%) of 577 individuals with insomnia and in 46 (8.0%) of
577 controls (OR = 1.62; 95% CI: 1.09–2.39; *P* = 0.01). Men with insomnia had a higher (16/
73: 21.9%) seroprevalence of *T. gondii* infection than men without insomnia (5/73: 6.8%)
(OR: 3.81; 95% CI: 1.31–11.06; *P* = 0.009). The rate of high (>150 IU/ml) anti-*T. gondii* IgG
antibody levels in cases was higher than the one in controls (OR = 2.21; 95% CI: 1.13–4.31;
*P* = 0.01). Men with insomnia had a higher (8/73: 11.0%) rate of high anti-*T. gondii* IgG anti-
body levels than men without insomnia (0/73: 0.0%) (*P* = 0.006). The rate of high anti-*T.
gondii* IgG antibody levels in cases >50 years old (11/180: 6.1%) was higher than that (3/
180: 1.7%) in controls of the same age group (OR: 3.84; 95% CI: 1.05–14.00; *P* = 0.05). No
difference in the rate of IgM seropositivity between cases and controls was found (OR =
1.33; 95% CI: 0.57–3.11; *P* = 0.50). Results of this seroepidemiology study suggest that
infection with *T. gondii* is associated with insomnia. Men older than 50 years with *T. gondii*
exposure might be prone to insomnia. Further research to confirm the association between
seropositivity and serointensity to *T. gondii* and insomnia is needed.

## Introduction

*Toxoplasma gondii* (*T. gondii*) is an obligate intracellular parasite [1]. Toxoplasmosis, the dis-
ease caused by *T. gondii*, is a zoonosis with medical and veterinary importance worldwide [2].
*T. gondii* infects one third of the global human population [3]. Foodborne transmission of *T.
gondii* in humans occurs mainly by eating undercooked meat, especially pork, lamb, and wild
game meat from *T. gondii*-infected animals, and ingestion of raw fruits and vegetables

**Funding:** This study was financially supported by Secretary of Public Education, Mexico (Grant No. DSA/103.5/14/11311). The funders had no role in study design, data collection and analysis, decision to publish or preparation of the manuscript.

**Competing interests:** The authors have declared that no competing interests exist.

contaminated with soil containing cat feces [4]. Transmission of *T. gondii* may also occur by blood transfusion, tissue transplants, or ingestion of unpasteurized milk [5]. In addition, *T. gondii* can be transmitted vertically during pregnancy [6]. Infections with *T. gondii* may result in a clinical spectrum ranging from a completely asymptomatic infection to multi-organ involvement [7]. Some patients with toxoplasmosis present cervical lymphadenopathy or ocular disease, whereas a reactivation of latent disease in immunocompromised patients can cause life-threatening encephalitis [8]. Chronic toxoplasmosis might be associated with a vast array of neuropsychiatric symptoms [9]. Seroprevalence of *T. gondii* infection has been linked to mixed anxiety and depressive disorder [10], schizophrenia [11, 12], depression [13, 14], and suicide behavior [14, 15]. Few studies have shown a link between *T. gondii* infection and insomnia. However, results of such studies are controversial. In a study of students, a positive association between seropositivity to *T. gondii* and insomnia was found in women but not in men [16]. In two further studies, researchers found that *T. gondii* IgG seropositivity and serointensity were not associated with insomnia [17, 18]. *T. gondii* can disseminate to brain in infected hosts [3], and we hypothesize that *T. gondii* can involve brain structures related with sleep leading to sleep problems as insomnia. Several neurotransmitters are involved in sleep including for instance, dopamine and gamma-aminobutyric acid (GABA) [19], and *T. gondii* infections impact on these two neurotransmitters. *T. gondii* infections alter the dopamine metabolism [20], and GABAergic synapses and signaling in the central nervous system [21]. Therefore, this study was aimed to determine the association between *T. gondii* exposure and insomnia in a sample of people in Durango, Mexico.

## Materials and methods

### Study design and study population

Through an age- and gender-matched case-control study, 577 people with insomnia (cases) and 577 people without insomnia (controls) were examined for the presence of *T. gondii* exposure. This study was performed in the northern Mexican state of Durango from 2014 to 2019. Cases and controls were individuals attending public health care institutions for medical consultations or laboratory tests. Inclusion criteria for cases were people with insomnia, aged ≥18 years, with informed consent. Gender, occupation, and socioeconomic status were not restrictive criteria for enrollment of cases or controls. The Diagnostic and Statistical Manual of Mental Disorders, Fifth edition (DSM-5) defines insomnia as a predominant complain of dissatisfaction with sleep quantity or quality, associated with one (or more) of the following symptoms: difficulty initiating sleep, difficulty maintaining sleep, and early-morning awakening with inability to return to sleep [22]. Inclusion criteria for enrollment of controls were people without insomnia, aged ≥18 years, with informed consent. Of the 577 cases, 505 (87.5%) were women and 72 (12.5%) were men. They were 18–80 years old (mean: 43.63 ± 12.55). Whereas of the 577 controls, 503 (87.2%) were women and 74 (12.8%) were men. They were 18–81 years old (mean: 43.63 ± 12.56). There were no differences in age ($P = 0.99$) or gender ($P = 0.93$) among cases and controls.

### Laboratory tests

Five ml of venous blood was drawn from each participant. After blood clotting, blood samples were centrifuged. Serum was transferred into microtubes and kept at -20°C until assayed. All samples were analyzed for anti-*T. gondii* IgG antibodies using the commercially available enzyme-linked immunoassay "*Toxoplasma gondii* IgG" kit (Diagnostic Automation/Cortez Diagnostics, Inc., Woodland Hills, California. USA). This test allows detection and quantification of anti-*T. gondii* IgG antibodies. A cut-off of 8 IU/ml was used to determine seropositivity.

Samples with results just below 8 IU/ml (grey zone) or clearly lower were considered as negatives. Serum samples with anti-*T. gondii* IgG antibodies were further analyzed for detection of anti-*T. gondii* IgM antibodies by the commercially available enzyme-linked immunoassay "*Toxoplasma gondii* IgM" kit (Diagnostic Automation/Cortez Diagnostics, Inc.). Sera were analyzed every few months during the study period. Positive and negative controls were included in each run. All immunoassays were carried out as described in the manufacturer's instructions. Laboratory tests were performed blindly, the analyst did not have information about the history of insomnia in participants during the analysis of samples.

## Statistical analysis

Data analysis was performed using the software SPSS Statistics version 15. A sample size of 577 cases and 577 controls was calculated using the following parameters: a reference seroprevalence of 6.1% [23] as the expected frequency of exposure in controls, a two-sided confidence level of 95%, a power of 90%, a 1:1 proportion of cases and controls, and an odds ratio of 2.0. The student's t-test was used to assess age matching. To compare the frequencies among the groups the Pearson's chi-square test or the Fisher's exact test (when a value was below 5) were used. Multivariate analysis with adjustment by age, sex, education, and residence to further assess the association between *T. gondii* infection and insomnia was performed. Odd ratios (OR) and corresponding 95% confidence intervals (CI) were calculated. A statistical *P* value of $\leq$ 0.05 was considered significant for all comparisons.

## Ethical aspects

The Ethical Committees of the Institute of Security and Social Services for the State Workers and the General Hospital of the Secretary of Health in Durango City, Mexico approved this project (Approval No. 449/015). A written informed consent from each participant was obtained.

## Results

Anti-*T. gondii* IgG antibodies were found in 71 (12.3%) of 577 individuals with insomnia and in 46 (8.0%) of 577 controls. A statistically significant difference (OR = 1.62; 95% CI: 1.09–2.39; *P* = 0.01) in anti-*T. gondii* IgG seroprevalence between cases and controls was found. A stratification by age and sex and seroprevalence of *T. gondii* infection in individuals with and without insomnia is shown in Table 1. Stratification by sex showed that men with insomnia had a significantly higher (16/73: 21.9%) seroprevalence of *T. gondii* infection than men without insomnia (5/73: 6.8%) (OR: 3.81; 95% CI: 1.31–11.06; *P* = 0.009). Whereas stratification by age groups showed that seroprevalence of *T. gondii* infection in cases was similar to controls regardless their age groups. Of the 71 anti-*T. gondii* IgG positive cases, 28 (39.4%) had anti-*T. gondii* IgG antibody levels higher than 150 IU/ml, 6 (8.5%) between 100 IU/ml and 150 IU/ml, and 37 (52.1%) between 8 to 99 IU/ml. On the other hand, of the 46 anti-*T. gondii* IgG positive controls, 13 (28.2%) had anti-*T. gondii* IgG antibody levels higher than 150 IU/ml, 5 (10.9%) between 100 IU/ml and 150 IU/ml, and 28 (60.9%) between 8 to 99 IU/ml. A statistically significant difference (OR = 2.21; 95% CI: 1.13–4.31; *P* = 0.01) in the frequency of individuals with high (>150 IU/ml) anti-*T. gondii* IgG antibody levels between cases and controls was found. A stratification by sex and age groups and the rates of high (>150 IU/ml) anti-*T. gondii* IgG antibody levels in cases and controls is shown in Table 2. Stratification by sex showed that men with insomnia had a significantly higher (8/73: 11.0%) rate of high anti-*T. gondii* IgG antibody levels than men without insomnia (0/73: 0.0%) (*P* = 0.006). Whereas stratification by age groups showed that the rate of high anti-*T. gondii* IgG antibody levels in cases >50 years

**Table 1. Stratification by sociodemographic variables in cases and controls and IgG seropositivity rates to *T. gondii*.**

| | Cases | | | Controls | | | | 95% | |
| | | Seropositivity | | | Seropositivity | | | confidence | |
| | No. | to *T. gondii* | | No. | to *T. gondii* | | Odds | | *P* |
| Characteristics | tested | No. | % | tested | No. | % | ratio | interval | value |
|---|---|---|---|---|---|---|---|---|---|
| Sex | | | | | | | | | |
| Male | 73 | 16 | 21.9 | 73 | 5 | 6.8 | 3.81 | 1.31–11.06 | 0.009 |
| Female | 504 | 55 | 10.9 | 504 | 41 | 8.1 | 1.38 | 0.9–2.1 | 0.13 |
| Age (years) | | | | | | | | | |
| ≤30 | 99 | 9 | 9.1 | 99 | 5 | 5.1 | 1.80 | 0.6–5.8 | 0.26 |
| 31–50 | 298 | 37 | 12.4 | 298 | 27 | 9.1 | 1.42 | 0.8–2.4 | 0.18 |
| >50 | 180 | 25 | 13.9 | 180 | 14 | 7.8 | 1.91 | 0.9–3.8 | 0.06 |
| Educational level | | | | | | | | | |
| No education | 11 | 5 | 45.5 | 8 | 3 | 37.5 | 1.38 | 0.2–8.9 | 0.72 |
| 1 to 6 years | 149 | 24 | 16.1 | 104 | 15 | 14.4 | 1.13 | 0.5–2.2 | 0.71 |
| 7–12 years | 261 | 27 | 10.3 | 303 | 18 | 5.9 | 1.80 | 0.9–3.3 | 0.05 |
| >12 years | 154 | 15 | 9.7 | 157 | 9 | 5.7 | 1.77 | 0.7–4.1 | 0.18 |
| Residence area | | | | | | | | | |
| Urban | 362 | 32 | 8.8 | 467 | 28 | 6.0 | 1.5 | 0.8–2.5 | 0.11 |
| Suburban | 108 | 16 | 14.8 | 49 | 6 | 12.2 | 1.2 | 0.4–3.4 | 0.66 |
| Rural | 105 | 23 | 21.9 | 56 | 12 | 21.4 | 1.0 | 0.4–2.2 | 0.94 |
| Total | 577 | 71 | 12.3 | 577 | 46 | 8 | 1.62 | 1.0–2.3 | 0.01 |

old (11/180: 6.1%) was higher than that (3/180: 1.7%) in controls of the same age group (OR: 3.84; 95% CI: 1.05–14.00; *P* = 0.05). Multivariate analysis with adjustment by age, sex, education, and residence showed that *T. gondii* infection was positively associated with insomnia (OR = 1.57; 95% CI: 1.06–2.34; *P* = 0.02). Of the 71 anti-*T. gondii* IgG antibody seropositive cases, 21 (29.6%) were also positive for anti-*T. gondii* IgM antibodies. Whereas, of the 46 anti-*T. gondii* IgG antibody seropositive controls, 11 (23.9%) were also positive for anti-*T. gondii* IgM antibodies. No difference in the rate of IgM seropositivity between cases and controls was found (OR = 1.33; 95% CI: 0.57–3.11; *P* = 0.50).

The dataset of the study that includes all the data used to obtain the results and conclusions of the study is available (S1 File).

**Table 2. Stratification by sex and age in cases and controls and rates of high (≥150 IU/ml) anti-*T. gondii* IgG antibody levels.**

| | Cases | | | Controls | | | | | |
| | | ≥150 IU/ml | | | ≥150 IU/ml | | | | |
| | | anti-*T. gondii* | | | anti-*T. gondii* | | | | |
| | No. | IgG levels | | No. | IgG levels | | Odds | confidence | *P* |
| Characteristics | tested | No. | % | tested | No. | % | ratio | interval | value |
|---|---|---|---|---|---|---|---|---|---|
| Sex | | | | | | | | | |
| Male | 73 | 8 | 11.0 | 73 | 0 | 0.0 | – | — | 0.006 |
| Female | 504 | 20 | 4.0 | 504 | 13 | 2.6 | 1.56 | 0.76–3.17 | 0.21 |
| Age (years) | | | | | | | | | |
| ≤30 | 99 | 4 | 4.0 | 99 | 1 | 1.0 | 4.12 | 0.45–37.60 | 0.36 |
| 31–50 | 298 | 13 | 4.4 | 298 | 9 | 3.0 | 1.46 | 0.61–3.48 | 0.38 |
| >50 | 180 | 11 | 6.1 | 180 | 3 | 1.7 | 3.84 | 1.05–14.00 | 0.05 |
| All | 577 | 28 | 4.9 | 577 | 13 | 2.3 | 2.21 | 1.13–4.31 | 0.01 |

## Discussion

There are only few studies that have assessed the epidemiological link between *T. gondii* infection and insomnia. Results of such studies are controversial and there is currently scanty information as to state whether *T. gondii* exposure leads to insomnia. Therefore, in this age- and gender-matched case-control study we attempted to determine the association between *T. gondii* exposure and insomnia in a sample of people in the northern Mexican state of Durango. We observed a significantly higher seroprevalence of *T. gondii* infection in individuals with insomnia than in individuals without insomnia. In addition, we found a significantly higher rate of high anti-*T. gondii* IgG antibody levels in cases than in controls. Therefore, our results suggest that *T. gondii* seropositivity and serointensity are positively associated with insomnia. The association between *T. gondii* infection and insomnia remained significant after multivariable analysis with adjustment for age, sex, education, and residence. However, the association of *T. gondii* seropositivity and insomnia was observed in men but not in women, and the association between *T. gondii* serointensity and insomnia was found only in men and in individuals aged >50 years. The positive association between *T. gondii* seropositivity and insomnia found in our study contrast with that found in a previous study of 200 students in Iran, where investigators observed a positive association between *T. gondii* seroprevalence and insomnia in women but not in men [16]. It is not clear why the association between seropositivity and insomnia was found only in men in our study. It is unknown whether men are more vulnerable than women to the clinical effects of *T. gondii*. It is possible that factors associated with *T. gondii* exposure in men were different from those in women in our study. Further research to determine the association between sociodemographic, clinical, and behavioral factors and *T. gondii* exposure in men with insomnia is needed. On the other hand, results of our study also contrast with the findings of three more studies. In a study of 311 older adults recruited in a psychiatric institute and clinic in Pittsburg, USA, researchers found no association between insomnia and *T. gondii* seroprevalence and serointensity [17]. In a study of 2031 Old Order Amish individuals recruited in Lancaster County, USA, researchers found no association between *T. gondii* seropositivity and serointensity and insomnia [18]. However, a secondary analysis identified, after adjustment by age group, a statistical trend towards shorter sleep duration in seropositive men [18]. In a survey of 833 Old Order Amish participants, *T. gondii* seropositives reported less sleep problems and less daytime problems due to poor sleep, and higher *T. gondii* titers were associated with longer sleep duration, earlier bedtime, and earlier mid-sleep time [24]. Difference in the association between *T. gondii* exposure and insomnia among the studies might be due to differences in the characteristics of the study populations. We studied people enrolled at several health care institutions whereas researchers of other studies enrolled students [16], older adults [17], and Old Order Amish individuals [18, 24]. In addition, there are differences in the sample size and study design among the studies. We used an age- and gender-matched case-control study design and studied 577 cases and 577 controls whereas in another study 180 people suffering from insomnia and 131 normal sleeping participants with no differences in age, gender and race were studied [17]. The study of students was cross-sectional and had 200 participants [16]; and the studies of Old Order Amish were cross-sectional and included between 833 [24] and 2031 participants [18]. Risk factors for *T. gondii* exposure and host factors might be different among the study populations. In addition, differences in the virulence of *T. gondii* strains among infected hosts might exist. In an experimental mouse model of chronic *T. gondii* infection, investigators found that infected mice exhibited chronic sleep–wake alterations over months, including a marked increase in time spent awake [25]. Further research to determine the association between insomnia and *T. gondii* seropositivity and serointensity are needed. The presence of anti-*T. gondii* IgM antibody was

determined only in anti-*T. gondii* IgG positive samples. The presence of IgM without IgG is not a reliable result because enzyme immune assays for anti-*T. gondii* IgM antibodies have high numbers of false-positive results [26]. We did not obtain information about specific sleep variables. Studies to determine an association between *T. gondii* infection and specific clinical variables of insomnia are needed. We did not screen for neuropsychiatric or neurodegenerative disorders among participants. Further studies with information about sleep clinical variables and the presence of neuropsychiatric or neurodegenerative disorders among participants are needed. Results of our study do not mean that *T. gondii* infection leads to insomnia. Only longitudinal studies may help to determine the direction of the association.

## Conclusions

Results of this age- and gender-matched case-control study suggest that infection with *T. gondii* is associated with insomnia. Men older than 50 years with *T. gondii* exposure might be prone to insomnia. Further research to confirm the association between seropositivity and serointensity to *T. gondii* and insomnia is needed.

## Supporting information

**S1 File. Date set of the study.**
(XLS)

## Author Contributions

**Conceptualization:** Cosme Alvarado-Esquivel, Elizabeth Rábago-Sánchez.

**Data curation:** Sergio Estrada-Martínez, Alma Rosa Pérez-Álamos, Agar Ramos-Nevárez, Raquel Vaquera-Enríquez, Carlos Alberto Guido-Arreola, Leandro Saenz-Soto.

**Formal analysis:** Cosme Alvarado-Esquivel, Sergio Estrada-Martínez, Alma Rosa Pérez-Álamos, Karina Botello-Calderón, Ángel Osvaldo Alvarado-Félix, Gustavo Alexis Alvarado-Félix.

**Funding acquisition:** Cosme Alvarado-Esquivel.

**Investigation:** Cosme Alvarado-Esquivel, Agar Ramos-Nevárez, Carlos Alberto Guido-Arreola, Leandro Saenz-Soto.

**Methodology:** Cosme Alvarado-Esquivel, Raquel Vaquera-Enríquez, Antonio Sifuentes-Álvarez, Elizabeth Rábago-Sánchez.

**Project administration:** Cosme Alvarado-Esquivel.

**Software:** Sergio Estrada-Martínez, Alma Rosa Pérez-Álamos.

**Writing – original draft:** Cosme Alvarado-Esquivel.

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
