## [Decision Letter · Decision Letter 0]

19 Apr 2022

PONE-D-22-07752Toxoplasma gondii infection and insomnia: a case control seroprevalence studyPLOS ONE

Dear Dr. Alvarado-Esquivel,

Thank you for submitting your manuscript to PLOS ONE. After careful consideration, we feel that it has merit but does not fully meet PLOS ONE’s publication criteria as it currently stands. Therefore, we invite you to submit a revised version of the manuscript that addresses the points raised during the review process.

ACADEMIC EDITOR:Please submit your revised manuscript by Jun 03 2022 11:59PM. If you will need more time than this to complete your revisions, please reply to this message or contact the journal office at plosone@plos.org. Please include the following items when submitting your revised manuscript:A rebuttal letter that responds to each point raised by the academic editor and reviewer(s). You should upload this letter as a separate file labeled 'Response to Reviewers'.A marked-up copy of your manuscript that highlights changes made to the original version. You should upload this as a separate file labeled 'Revised Manuscript with Track Changes'.An unmarked version of your revised paper without tracked changes. You should upload this as a separate file labeled 'Manuscript'.

We look forward to receiving your revised manuscript.

Kind regards,

Masoud Foroutan, Ph.D; Assistant Professor

Academic Editor

PLOS ONE

Journal Requirements:

[No authors have competing interests.]

Reviewers' comments:

Reviewer's Responses to Questions

**Comments to the Author**

1. Is the manuscript technically sound, and do the data support the conclusions?

Reviewer #1: Partly

Reviewer #2: Partly

Reviewer #3: Yes

2. Has the statistical analysis been performed appropriately and rigorously? 

Reviewer #1: Yes

Reviewer #2: No

Reviewer #3: Yes

3. Have the authors made all data underlying the findings in their manuscript fully available?

Reviewer #1: No

Reviewer #2: Yes

Reviewer #3: Yes

4. Is the manuscript presented in an intelligible fashion and written in standard English?

Reviewer #1: Yes

Reviewer #2: Yes

Reviewer #3: Yes

5. Review Comments to the Author

Reviewer #1: The topic is interesting but the manuscript needs major revision.

The approval ethical number needs to add.

The authors only checked the serum samples with anti-T. gondii IgG antibodies for analyze and detection of anti-T. gondii IgM antibodies so may be lost the samples with IgM positive and IgG negative.

The data of table 1 is the positive of T. gondii is related to IgG or IgM results.

There is not any tables about IgM results.

Is T. gondii related to the kind of symptoms: 1-difficulty initiating sleep, 2- difficulty maintaining sleep and 3- early-morning awakening with inability to return to sleep, or no?

The duration of this study from 2014-2019 is very wide period and this may be affected the age grouping.

All sera that collected during 5 years were evaluated for IgG and IgM against T. gondii in the same time or no.

Reviewer #2: The article submitted for publication by C . Alvarado-Esquivel and colleagues deals with the association between Toxoplasma infection and insomnia. It reports the findings of a case control study performed on 577 patients with insomnia and 577 control subjects in Mexico, and concludes to the presence of association.

Introduction

l.88 : « responsible » there is no link of causality that has been clearly demonstrated, only statistical association

l90-95 : As mentioned by the authors several studies have already looked into this association with conflicting results, however article by Corona C should be cited and discussed (Toxoplasma gondii IgG associations with sleep-wake problems, sleep duration and timing, Pteridines. 2019 Feb; 30(1): 1–9.). Furthermore we believe that the only fundamental study in a mice model should also be included and discussed (Dupont et al, Chronic Toxoplasma gondii infection and sleep-wake alterations in mice, CNS Neurosci Ther. 2021 Aug;27(8):895-907).

Methods

The heterogeneity of symptoms associated with insomnia should also be commented on in the discussion.

Selection of controls: it is known that T. gondii seroprevalence is strongly influenced by education or residence? Thus are controls similar to cases on these parameters? If no information available, this should be discussed. Furthermore it is known that sleep is an index of brain functions as sleep alteration are usually observed in several neuropsychiatric or neurodegenerative disorders even before clinical neurological manifestations. If controls/cases were selected when attending medical institution, did these patients were screened for these disorders associated with sleep alterations? this should also be discussed.

Age groups are not described in the methods, we believe that analyzing the effects of age as a continuous variable should be better thant categorizing by arbitrary age groups.

Laboratory tests

Were the analyses performed blindly as regards to the staus of cases/controls ? Was there a grey zone in the assay used ? if so how was it further processed ?

Discussion

Reference 18 does not conclude to an association however, there is statistical trend toward shorter sleep duration in seropositive men. This should be added to the discussion, in line with results from Dupont et al, Chronic Toxoplasma gondii infection and sleep-wake alterations in mice, CNS Neurosci Ther. 2021 Aug;27(8):895-907.

Furthermore, it should be clearly stated that this study does not allow to conclude that T. gondii leads to insomnia. Statistics do only point towards an association without precluding the direction of the association, all the most if patients were not selected as free of neuropsychiatric or degenerative disorders. Cinetics is of utmost importance and cinetics cannot be performed when using only one serum.

Reviewer #3: Dear respect Editor:

The manuscript entitled: “Toxoplasma gondii infection and insomnia: a case control seroprevalence study” addresses a highly interesting topic. The study presents remarkable information. The methodology and design of this work is appropriate. I suggest that the authors could enhance the usefulness of this work for the community if they redesign the tables.

6. PLOS authors have the option to publish the peer review history of their article (what does this mean?). If published, this will include your full peer review and any attached files.

Reviewer #1: No

Reviewer #2: No

Reviewer #3: No

---

## [Author Response · Author response to Decision Letter 0]

12 May 2022

Durango, Dgo. Mexico. May 4, 2022.

Dear Editor,

Please find attached a revised version of our manuscript that has been modified according to the reviewers’ comments. In addition, please find below our response to each of the reviewers’ comments on a point-by-point basis. The revised manuscript meets PLOS ONE’s style requirements. The authors have declared that no competing interests exist. The dataset of the study that includes all the data used to obtain the results and conclusions of the study is available (Supplementary file 1).

We appreciate the valuable comments of the reviewers and we hope the revised manuscript may have more success for publication in the journal Plos One.

Kind regards,

Dr. Cosme Alvarado-Esquivel.

Laboratorio de Investigación Biomédica

Facultad de Medicina y Nutrición

Avenida Universidad S/N.

34000 Durango, Dgo. Mexico.

Tel/Fax.: 0052 618 8 271200

Email: alvaradocosme@yahoo.com

 

RESPONSE TO THE REVIEWERS’ COMMENTS

Journal Requirements:

The manuscript was modified according to the PLOS ONE’s style requirements. 

2. If you have no competing interests, please state "The authors have declared that no competing interests exist.", as detailed online in our guide for authors at http://journals.plos.org/plosone/s/submit-now

The cover letter includes the following statement: “The authors have declared that no competing interests exist”.

Thank you for your valuable comments for improving our manuscript.

Reviewers' comments:

Reviewer #1: 

1. The approval ethical number needs to add.

The approval ethical number was added (line 129).

2. The authors only checked the serum samples with anti-T. gondii IgG antibodies for analyze and detection of anti-T. gondii IgM antibodies so may be lost the samples with IgM positive and IgG negative.

The presence of anti-T. gondii IgM antibody was determined only in anti-T. gondii IgG positive samples. The presence of IgM without IgG is not a reliable result because enzyme immune assays for anti-T. gondii IgM antibodies have high numbers of false-positive results. This information was added to the Discussion section (lines 221-224). 

3. The data of table 1 is the positive of T. gondii is related to IgG or IgM results.

It is related to IgG antibodies. This information was added to the Title of Table 1.

4. There is not any tables about IgM results.

Since there was no difference in the frequency of IgM antibodies among groups, no further analysis of IgM results in a Table was performed.

5. Is T. gondii related to the kind of symptoms: 1-difficulty initiating sleep, 2- difficulty maintaining sleep and 3- early-morning awakening with inability to return to sleep, or no?

We did not obtain this information. Studies to determine an association between T. gondii infection and specific symptoms of insomnia are needed. This information was added to the Discussion section (lines 224-226). 

6. The duration of this study from 2014-2019 is very wide period and this may be affected the age grouping.

We further evaluate the association between T. gondii seropositivity and insomnia by adjustment for age. This information was added to the Materials and methods section (lines 121-123), Results section (lines 155-157), and Discussion section (lines 183-185). 

7. All sera that collected during 5 years were evaluated for IgG and IgM against T. gondii in the same time or no.

Sera were analyzed every few months during the study period. This information was added to the Materials and methods section (line 108). 

Thank you for your valuable comments for improving our manuscript.

Reviewer #2: 

1. Introduction

l.88 : « responsible » there is no link of causality that has been clearly demonstrated, only statistical association

The word “responsible” was deleted. The sentence was rewritten as follows: ”Chronic toxoplasmosis might be associated with a…” (line 62).

2. l90-95: As mentioned by the authors several studies have already looked into this association with conflicting results, however article by Corona C should be cited and discussed (Toxoplasma gondii IgG associations with sleep-wake problems, sleep duration and timing, Pteridines. 2019 Feb; 30(1): 1–9.). Furthermore we believe that the only fundamental study in a mice model should also be included and discussed (Dupont et al, Chronic Toxoplasma gondii infection and sleep-wake alterations in mice, CNS Neurosci Ther. 2021 Aug;27(8):895-907).

Our results were contrasted with those reported by Corona et al, and Dupont et al, (lines 203-206, and lines 218-220, respectively). 

3. Methods

The heterogeneity of symptoms associated with insomnia should also be commented on in the discussion.

We did not obtain this information. Studies to determine an association between T. gondii infection and specific symptoms of insomnia are needed. This information was added to the Discussion section (lines 224-226). 

4. Selection of controls: it is known that T. gondii seroprevalence is strongly influenced by education or residence? Thus are controls similar to cases on these parameters? If no information available, this should be discussed. 

Information about education and residence in participants was added to Table 1. We further evaluate the association between T. gondii seropositivity and insomnia by adjustment for age, education and residence. This information was added to the Materials and methods section (lines 121-123), Results section (lines 155-157), and Discussion section (lines 183-185). 

5. Furthermore it is known that sleep is an index of brain functions as sleep alteration are usually observed in several neuropsychiatric or neurodegenerative disorders even before clinical neurological manifestations. If controls/cases were selected when attending medical institution, did these patients were screened for these disorders associated with sleep alterations? this should also be discussed.

This was a limitation of the study. We did not screen for neuropsychiatric or neurodegenerative disorders among participants. Further studies with information about sleep clinical variables and the presence of neuropsychiatric or neurodegenerative disorders among participants are needed (lines 226-229). 

6. Age groups are not described in the methods, we believe that analyzing the effects of age as a continuous variable should be better thant categorizing by arbitrary age groups.

We further evaluated the association between T. gondii seropositivity and insomnia by adjustment for age. This information was added to the Materials and methods section (lines 121-123), Results section (lines 155-157), and Discussion section (lines 183-185). 

7. Laboratory tests

Were the analyses performed blindly as regards to the staus of cases/controls ? 

Laboratory tests were performed blindly, the analyst did not have information about the history of insomnia in participants during the analysis of samples. This information was added to the Materials and methods section (lines 110-112).

8. Was there a grey zone in the assay used ? if so how was it further processed ?

Samples with results just below 8 IU/ml (grey zone) or clearly lower were considered as negatives (lines 104-105).

9. Discussion

Reference 18 does not conclude to an association however, there is statistical trend toward shorter sleep duration in seropositive men. This should be added to the discussion, in line with results from Dupont et al, Chronic Toxoplasma gondii infection and sleep-wake alterations in mice, CNS Neurosci Ther. 2021 Aug;27(8):895-907.

Information about the secondary analysis (reference 18) that showed a trend toward shorter sleep duration in seropositive men was added in the Discussion section (lines 201-203).

10. Furthermore, it should be clearly stated that this study does not allow to conclude that T. gondii leads to insomnia. Statistics do only point towards an association without precluding the direction of the association, all the most if patients were not selected as free of neuropsychiatric or degenerative disorders. Cinetics is of utmost importance and cinetics cannot be performed when using only one serum.

We added the following information: Results of our study do not mean that T. gondii infection leads to insomnia. Only longitudinal studies may help to determine the direction of the association (lines 229-231). 

Thank you for your valuable comments for improving our manuscript.

Reviewer #3: 

1. I suggest that the authors could enhance the usefulness of this work for the community if they redesign the tables.

Table 1 was modified. Information about sociodemographic variables was added.

Thank you for your valuable comments for improving our manuscript.

---

## [Decision Letter · Decision Letter 1]

24 May 2022

Toxoplasma gondii infection and insomnia: a case control seroprevalence study

PONE-D-22-07752R1

Dear Dr. Alvarado-Esquivel,

We’re pleased to inform you that your manuscript has been judged scientifically suitable for publication and will be formally accepted for publication once it meets all outstanding technical requirements.

Kind regards,

Masoud Foroutan, Ph.D; Assistant Professor

Academic Editor

PLOS ONE

Additional Editor Comments (optional):

Reviewers' comments:

Reviewer's Responses to Questions

**Comments to the Author**

1. If the authors have adequately addressed your comments raised in a previous round of review and you feel that this manuscript is now acceptable for publication, you may indicate that here to bypass the “Comments to the Author” section, enter your conflict of interest statement in the “Confidential to Editor” section, and submit your "Accept" recommendation.

Reviewer #1: All comments have been addressed

Reviewer #2: All comments have been addressed

2. Is the manuscript technically sound, and do the data support the conclusions?

Reviewer #1: Yes

Reviewer #2: (No Response)

3. Has the statistical analysis been performed appropriately and rigorously? 

Reviewer #1: Yes

Reviewer #2: (No Response)

4. Have the authors made all data underlying the findings in their manuscript fully available?

Reviewer #1: No

Reviewer #2: (No Response)

5. Is the manuscript presented in an intelligible fashion and written in standard English?

Reviewer #1: Yes

Reviewer #2: (No Response)

6. Review Comments to the Author

Reviewer #1: The authors revised the manuscript according to reviewers comments. The limitations in this work need more discussed in the discussion.

Reviewer #2: (No Response)

7. PLOS authors have the option to publish the peer review history of their article (what does this mean?). If published, this will include your full peer review and any attached files.

Reviewer #1: No

Reviewer #2: No

---

## [Editor Report · Acceptance letter]

31 May 2022

PONE-D-22-07752R1 

*Toxoplasma gondii* infection and insomnia: a case control seroprevalence study 

Dear Dr. Alvarado-Esquivel:

I'm pleased to inform you that your manuscript has been deemed suitable for publication in PLOS ONE. Congratulations! Your manuscript is now with our production department. 

Kind regards, 

on behalf of

Dr. Masoud Foroutan 

Academic Editor

PLOS ONE